# Message Passing-Based Inference for Time-Varying Autoregressive Models

**DOI:** 10.3390/e23060683

**Published:** 2021-05-28

**Authors:** Albert Podusenko, Wouter M. Kouw, Bert de Vries

**Affiliations:** 1Department of Electrical Engineering, Eindhoven University of Technology, 5612 AZ Eindhoven, The Netherlands; w.m.kouw@tue.nl (W.M.K.); bdevries@gnresound.com (B.d.V.); 2GN Hearing, JF Kennedylaan 2, 5612 AB Eindhoven, The Netherlands

**Keywords:** Bayesian inference, free energy, factor graph, hybrid message passing, model selection, non-stationary systems, probabilistic graphical models

## Abstract

Time-varying autoregressive (TVAR) models are widely used for modeling of non-stationary signals. Unfortunately, online joint adaptation of both states and parameters in these models remains a challenge. In this paper, we represent the TVAR model by a factor graph and solve the inference problem by automated message passing-based inference for states and parameters. We derive structured variational update rules for a composite “AR node” with probabilistic observations that can be used as a plug-in module in hierarchical models, for example, to model the time-varying behavior of the hyper-parameters of a time-varying AR model. Our method includes tracking of variational free energy (FE) as a Bayesian measure of TVAR model performance. The proposed methods are verified on a synthetic data set and validated on real-world data from temperature modeling and speech enhancement tasks.

## 1. Introduction

Autoregressive (AR) models are capable of describing a wide range of time series patterns [1,2]. The extension to Time-Varying AR (TVAR) models, where the AR coefficients are allowed to vary over time, supports tracking of non-stationary signals. TVAR models have been successfully applied to a wide range of applications, including speech signal processing [3,4,5], signature verification [6], cardiovascular response modeling [7], acoustic signature recognition of vehicles [8], radar signal processing [9], and EEG analysis [10,11].

The realization of TVAR models in practice often poses some computational issues. For many applications, such as speech processing in a hearing aid, both a low computational load and high modeling accuracy are essential.

The problem of parameter tracking in TVAR models has been extensively explored in a non-Bayesian setting. For example, ref. [12] employs over-determined modified Yule-Walker equations and [13] applies the covariance method to track the parameters in a TVAR model. In [14], expressions for the mean vector and covariance matrix of TVAR model coefficients are derived and [15] uses wavelets for TVAR model identification. Essentially, all these approaches provide maximum likelihood estimates of coefficients for TVAR models without measurement noise. In [16], autoregressive parameters were estimated from noisy observations by using a recursive least-squares adaptive filter.

We take a Bayesian approach since we are also interested in tracking Bayesian evidence (or an approximation thereof) as a model performance measure. Bayesian evidence can be used to track the optimal AR model order or more generally, to compare the performance of a TVAR model to an alternative model. To date, Bayesian parameter tracking in AR models has mostly been achieved by Monte Carlo sampling methods [17,18,19,20,21]. Sampling-based inference is highly accurate, but it is very often computationally too expensive for real-time processing on wearable devices such as hearing aids, smartwatches, etc.

In this paper, we develop a low-complexity variational message passing-based (VMP) realization for tracking of states, parameters and free energy (an upper bound on Bayesian evidence) in TVAR models. All update formulas are closed-form and the complete inference process can easily be realized.

VMP is a low-complexity distributed message passing-based realization of variational Bayesian inference on a factor graph representation of the model [22,23]. Previous work on message passing-based inference for AR models include [24], but their work describes maximum likelihood estimation and therefore does not track proper posteriors and free energy. In [25], variational inference is employed to estimate the parameters of a multivariate AR model, but their work does not take advantage of the factor graph representation.

The factor graph representation that we employ in this paper provides some distinct advantages from other works on inference in TVAR models. First, a factor graph formulation is by definition completely modular and supports re-using the derived TVAR inference equations as a plug-in module in other factor graph-based models. In particular, since we allow for measurement noise in the TVAR model specification, the proposed TVAR factor can easily be used as a latent module at any level in hierarchical dynamical models. Moreover, due to the modularity, VMP update rules can easily be mixed with different update schemes such as belief propagation and expectation [26,27] in other modules, leading to hybrid message passing schemes for efficient inference in complex models. We have implemented the TVAR model in the open source and freely available factor graph toolbox ForneyLab [28].

The rest of this paper is organized as follows. In Section 2, we specify the TVAR model as a probabilistic state space model and define the inference tasks that relate to tracking of states, parameters, and Bayesian evidence. After a short discussion on the merits of using Bayesian evidence as a model performance criterion (Section 3.1), we formulate Bayesian inference in the TVAR model as a set of sequential prediction-correction processes (Section 3.2). We will realize these processes as VMP update rules and proceed with a short review of Forney-style factor graphs and message passing in Section 4. Then, in Section 5, the VMP equations are worked out for the TVAR model and summarized in Table 1. Section 6 discusses a verification experiment on a synthetic data set and applications of the proposed TVAR model to temperature prediction and speech enhancement problems. Full derivations of the closed-form VMP update rules are presented in Appendix A.

## 2. Model Specification and Problem Definition

In this section, we first specify TVAR as a state-space model. This is followed by an inference problem formulation.

### 2.1. Model Specification

A TVAR model is specified as
(1a)$$\begin{array}{cc} \; & \theta_{m,t}∼N\left(\theta_{m,t-1},\omega \right) \end{array}$$
(1b)$$\begin{array}{cc} \; & x_{t}∼N\left(\sum_{m=1}^{M}\theta_{m,t}x_{t-m},\gamma^{-1}\right) \end{array}$$
(1c)$$\begin{array}{cc} \; & y_{t}∼N\left(x_{t},\tau \right)\;, \end{array}$$
where $y_{t}\in R$, $x_{t}\in R$ and $\theta_{m,t}\in R$ represent the the observation, state and parameters at time *t*, respectively. *M* denotes the order of the AR model. As a notational convention, we use N(μ,Σ) to denote a Gaussian distribution with mean μ and co-variance matrix Σ. We can re-write (1) in state-space form as
(2a)$$\begin{array}{cc} \theta_{t} & ∼N(\theta_{t-1},\omega I_{M}) \end{array}$$
(2b)$$\begin{array}{cc} x_{t} & ∼N\left(A\left(\theta_{t}\right)x_{t-1},V\left(\gamma \right)\right) \end{array}$$
(2c)$$\begin{array}{cc} y_{t} & ∼N(c^{⊺}x_{t},\tau )\;, \end{array}$$
where $\theta_{t}={\left(\theta_{m,t},\theta_{m-1,t},\cdots ,\theta_{m-M,t}\right)}^{⊺}$, $x_{t}={\left(x_{t},x_{t-1},\cdots ,x_{t-M+1}\right)}^{⊺}$, $c={\left(1,0,\ldots ,0\right)}^{⊺}$ is an *M*-dimensional unit vector, $V\left(\gamma \right)=\left(1/\gamma \right)cc^{⊺}$, and
(3)$$A\left(\theta \right)=\left[\begin{array}{cc} \;\theta^{⊺}\\ I_{M-1} & 0\;. \end{array}\right]$$

Technically, a TVAR model usually assumes τ=0 indicating that there is no measurement noise. Note that the presence of measurement noise in (2c) “hides” the states $x_{t}$ in the generative model (2) from the observation sequence $y_{t}$, yielding a latent TVAR. We add measurement noise explicitly, so the model is able to accept information from likelihood functions that are not constrained to be delta functions with hard observations. As a result, the AR model that we define here can be used at any level in deep hierarchical structures such as [29] as a plug-in module.

In a time-invariant AR model, θ are part of the parameters of the system. In a time-varying AR model, we consider $\theta_{t}$ and $x_{t}$ together the set of time-varying states. The parameters of the TVAR model are $\left\{\theta_{0},x_{0},\omega ,\gamma ,\tau \right\}$.

At the heart of the TVAR model is the transition model ([Disp-formula FD2b-entropy-23-00683]), where $A(\theta_{t})$ is a companion matrix with AR coefficients. The multiplication $A\left(\theta \right)x_{t-1}$ performs two operations: a dot product $\theta_{t}^{⊺}x_{t-1}$ and a vector shift of $x_{t-1}$ by one time step. The latter operation can be interpreted as bookkeeping, as it shifts each entry of $x_{t-1}$ one position down and discards $x_{t-M}$.

### 2.2. Problem Definition

For a given time series $y=(y_{1},y_{2},\ldots ,y_{T})$, we are firstly interested in recursively updating posteriors for the states $p(x_{t}|y_{1:l})$ and $p(\theta_{t}|y_{1:l})$. In this context, prediction, filtering and smoothing are recovered for l<t, l=t and l>t, respectively.

Furthermore, we are interested in computing posteriors for the parameters $p(\theta_{0}|y)$, $p(x_{0}|y)$, p(ω|y), p(γ|y) and p(τ|y).

Finally, we are interested in scoring the performance of a proposed TVAR model *m* with specified priors for the parameters. In this paper, we take a full Bayesian approach and select Bayesian evidence p(y|m) as the performance criterion. Section 3.1 discusses the merits of Bayesian evidence as a model performance criterion.

## 3. Inference in TVAR Models

In this section, we first discuss some of the merits of using Bayesian evidence as a model performance criterion. This is followed by an exposition of how to compute Bayesian evidence and the desired posteriors in the TVAR model.

### 3.1. Bayesian Evidence as a Model Performance Criterion

Consider a model *m* with parameters θ and observations *y*. Bayesian evidence p(y|m) is considered an excellent model performance criterion. Note the following decomposition [30]:(4)$$\begin{array}{cc} \log p(y|m) & =\log\frac{p(y|\theta ,m)p(\theta |m)}{p(\theta |y,m)}\;(\mathrm{use}\;\mathrm{Bayes}\;\mathrm{rule})\\ \; & =\int p\left(\theta |y,m\right)\cdot \underset{\log p(y|m)\;\mathrm{is}\;\mathrm{not}\;a\;\mathrm{function}\;\mathrm{of}\;\theta }{\underset{︸}{\log\frac{p(y|\theta ,m)p(\theta |m)}{p(\theta |y,m)}}}d\theta \\ \; & =\underset{\mathrm{data}\;\mathrm{fit}}{\underset{︸}{\int p(\theta |y,m)\log p(y|\theta ,m)d\theta }}-\underset{\mathrm{complexity}}{\underset{︸}{\int p\left(\theta |y,m\right)\log\frac{p(\theta |y,m)}{p(\theta |m)}d\theta }} \end{array}$$

The first term (data fit or sometimes called accuracy) measures how well the model predicts the data y, after having learned from the data. We want this term to be large (although only focusing on this term could lead to over-fitting). The second term (complexity) quantifies the amount of information that the model absorbed through learning by moving parameter beliefs from p(θ|m) to p(θ|y,m). To see this, note that the mutual information between two variables θ and y, which is defined as
$$I\left[\theta ;y\right]=\;\int \int \;p\left(\theta ,y\right)\log\frac{p(\theta |y)}{p(\theta )}d\theta dy\;,$$
can be interpreted as expected complexity. The complexity term regularizes the Bayesian learning process automatically. Preference for models with high Bayesian evidence implies a preference for models that get a good data fit without the need to learn much from the data set. These types of models are said to *generalize* well, since they can be applied to different data sets without specific adaptations for each data set. Therefore, Bayesian learning automatically leads to models that tend to generalize well.

Note that Bayesian evidence for a model *m*, given a full times series $y=(y_{1},y_{2},\ldots ,y_{T})$, can be computed by multiplication of the sample-based evidences:(5)$$p\left(y|m\right)=\prod_{t=1}^{T}p\left(y_{t}|y_{1:t-1},m\right)\;.$$

### 3.2. Inference as a Prediction-Correction Process

To illustrate the type of calculations that are needed for computing Bayesian model evidence and the posteriors for states and parameters, we now proceed to write out the needed calculations for the TVAR model in a filtering context.

Assume that at the beginning of time step *t*, we are given the state posteriors $q(x_{t-1}|y_{1:t-1})$, $q(\theta_{t-1}|y_{1:t-1})$. We will denote the inferred probabilities by q(·), in contrast to factors from the generative model that are written as p(·). We start the procedure by setting the state priors for the generative model at step *t* to the posteriors of the previous time step
(6)$$\begin{array}{cc} p(x_{t-1}|y_{1:t-1}) & :=q(x_{t-1}|y_{1:t-1}) \end{array}$$
(7)$$\begin{array}{cc} p(\theta_{t-1}|y_{1:t-1}) & :=q(\theta_{t-1}|y_{1:t-1}) \end{array}$$

Given a new observation $y_{t}$, we are now interested inferring the evidence $q(y_{t}|y_{t-1})$, and in inferring posteriors $q(x_{t}|y_{1:t})$ and $q(\theta_{t}|y_{1:t})$.

This involves a prediction (forward) pass through the system that leads to the evidence update, followed by a correction (backward) pass that updates the states. We work this out in detail below. For clarity of exposition, in this section we call $x_{t}$ states and $\theta_{t}$ parameters. Starting with the forward pass (from latent variables toward observation), we first compute a parameter prior predictive:(8)$$\underset{\begin{array}{c} \mathrm{parameter}\\ \mathrm{prior}\;\mathrm{predictive} \end{array}}{\underset{︸}{q(\theta_{t}|y_{1:t-1})}}=\int \underset{\begin{array}{c} \mathrm{parameter}\\ \mathrm{transition} \end{array}}{\underset{︸}{p(\theta_{t}|\theta_{t-1})}}\underset{\begin{array}{c} \mathrm{parameter}\\ \mathrm{prior} \end{array}}{\underset{︸}{p(\theta_{t-1}|y_{1:t-1})}}d\theta_{t-1}\;.$$

Then the prior predictive for the state transition becomes:(9)$$\underset{\begin{array}{c} \mathrm{state}\;\mathrm{transition}\\ \mathrm{prior}\;\mathrm{predictive} \end{array}}{\underset{︸}{q(x_{t}|x_{t-1},y_{1:t-1})}}=\int \underset{\mathrm{state}\;\mathrm{transition}}{\underset{︸}{p(x_{t}|x_{t-1},\theta_{t})}}\underset{\begin{array}{c} \mathrm{parameter}\\ \mathrm{prior}\;\mathrm{predictive} \end{array}}{\underset{︸}{q(\theta_{t}|y_{1:t-1})}}d\theta_{t}\;.$$

Note that the state transition prior predictive, due to its dependency on time-varying $\theta_{t}$, is a function of the observed data sequence. The state transition prior predictive can be used together with the state prior to inferring the state prior predictive:(10)$$\underset{\begin{array}{c} \mathrm{state}\\ \mathrm{prior}\;\mathrm{predictive} \end{array}}{\underset{︸}{q(x_{t}|y_{1:t-1})}}=\int \underset{\begin{array}{c} \mathrm{state}\;\mathrm{transition}\\ \mathrm{prior}\;\mathrm{predictive} \end{array}}{\underset{︸}{q(x_{t}|x_{t-1},y_{1:t-1})}}\underset{\mathrm{state}\;\mathrm{prior}}{\underset{︸}{p(x_{t-1}|y_{1:t-1})}}dx_{t-1}\;.$$

The evidence for model *m* that is provided by observation $y_{t}$ is then given by
(11)$$\underset{\mathrm{evidence}}{\underset{︸}{q(y_{t}|y_{1:t-1})}}=\int \underset{\begin{array}{c} \mathrm{state}\\ \mathrm{likelihood} \end{array}}{\underset{︸}{p(y_{t}|x_{t})}}\underset{\begin{array}{c} \mathrm{state}\;\mathrm{prior}\\ \mathrm{predictive} \end{array}}{\underset{︸}{q(x_{t}|y_{1:t-1})}}dx_{t}\;.$$

When $y_{t}$ has not yet been observed, $q(y_{t}|y_{1:t-1})$ is a prediction for $y_{t}$. After plugging in the observed value for $y_{t}$, the evidence is a scalar that scores how well the model performed in predicting $y_{t}$. As discussed in (Equation 5), the results $q(y_{t}|y_{1:t-1})$ for t=1,2,…,T in (Equation 11) can be used to score the model performance for a given time series $y=(y_{1},y_{2},\ldots ,y_{T})$. Note that to update the evidence, we need to integrate over all latent variables $\theta_{t-1}$, $\theta_{t}$, $x_{t-1}$ and $x_{t}$ (by (Equation 8)–(Equation 11)). In principle, this scheme needs to be extended with integration over the parameters ω, γ and τ.

Once we have inferred the evidence, we proceed by a backward corrective pass through the model to update the posterior over the latent variables given the new observation $y_{t}$. The state posterior can be updated by Bayes rule:(12)$$\underset{\mathrm{state}\;\mathrm{posterior}}{\underset{︸}{q(x_{t}|y_{1:t})}}=\frac{\overset{\begin{array}{c} \mathrm{state}\\ \mathrm{likelihood} \end{array}}{\overset{︷}{p(y_{t}|x_{t})}}\overset{\begin{array}{c} \mathrm{state}\;\mathrm{prior}\\ \mathrm{predictive} \end{array}}{\overset{︷}{q(x_{t}|y_{1:t-1})}}}{\underset{\mathrm{evidence}}{\underset{︸}{q(y_{t}|y_{1:t-1})}}}$$

Next, we need to compute a likelihood function for the parameters. Fortunately, we can re-use some intermediate results from the forward pass. The likelihood for the parameters is given by
(13)$$\underset{\begin{array}{c} \mathrm{parameter}\\ \mathrm{likelihood} \end{array}}{\underset{︸}{q(y_{t}|\theta_{t},y_{1:t-1})}}=\int \underset{\begin{array}{c} \mathrm{state}\\ \mathrm{likelihood} \end{array}}{\underset{︸}{p(y_{t}|x_{t})}}\underset{\begin{array}{c} \mathrm{state}\;\mathrm{prior}\\ \mathrm{predictive} \end{array}}{\underset{︸}{q(x_{t}|\theta_{t},y_{1:t-1})}}dx_{t}$$

The parameter posterior then follows from Bayes rule:(14)$$\underset{\mathrm{parameter}\;\mathrm{posterior}}{\underset{︸}{q(\theta_{t}|y_{1:t})}}=\frac{\overset{\begin{array}{c} \mathrm{parameter}\\ \mathrm{likelihood} \end{array}}{\overset{︷}{q(y_{t}|\theta_{t},y_{1:t-1})}}\overset{\begin{array}{c} \mathrm{parameter}\\ \mathrm{prior}\;\mathrm{predictive} \end{array}}{\overset{︷}{q(\theta_{t}|y_{1:t-1})}}}{\underset{\mathrm{evidence}}{\underset{︸}{q(y_{t}|y_{1:t-1})}}}$$

Equations (Equation 11), (Equation 12) and (Equation 14) contain the solutions to our inference task. Note that the evidence $q(y_{t}|y_{1:t-1})$ is needed to normalize the latent variable posteriors in (Equation 12) and (Equation 14). Moreover, while we integrate over the states by (Equation 11) to compute the evidence, (Equation 14) reveals that the evidence can alternatively be computed by integrating over the parameters through
(15)$$\underset{\mathrm{evidence}}{\underset{︸}{q(y_{t}|y_{1:t-1})}}=\int \underset{\begin{array}{c} \mathrm{parameter}\\ \mathrm{likelihood} \end{array}}{\underset{︸}{q(y_{t}|\theta_{t},y_{1:t-1})}}\underset{\begin{array}{c} \mathrm{parameter}\\ \mathrm{prior}\;\mathrm{predictive} \end{array}}{\underset{︸}{q(\theta_{t}|y_{1:t-1})}}d\theta_{t}\;.$$

This latter method of evidence computation may be useful if re-using (Equation 11) in (Equation 14) leads to numerical rounding issues.

Unfortunately, many of Equations (Equation 8) through (Equation 14) are not analytically tractable for the TVAR model. This happens due to (1) integration over large state spaces, (2) non-conjugate prior-posterior pairing, and (3) the absence of a closed-form solution for the evidence factor.

To overcome this challenge, we will perform inference by a hybrid message passing scheme in a factor graph. In the next section, we give a short review of Forney-Style Factor Graphs (FFG) and Message-Passing (MP) based inference techniques.

## 4. Factor Graphs and Message Passing-Based Inference

In this section, we make a brief introduction of Forney-Style Factor graph (FFG) and sum-product (SP) algorithm. After that we review the minimization of variational free energy and Variational Message Passing (VMP) algorithm.

### 4.1. Forney-Style Factor Graphs

A Forney-style Factor graph is a representation of a factorized function where the factors and variables are represented by nodes and edges, respectively. An edge is connected to a node if and only if the (edge) variable is an argument of the node function. In our work, we use FFGs to represent factorized probability distributions. FFGs provide both an attractive visualization of the model and a highly efficient and modular inference method based on message passing. An important component of the FFG representation is the equality node. If a variable *x* is shared between more than two nodes, then we introduce two auxiliary variables $x^{\prime }$ and $x^{\prime \prime }$ and use an equality node
(16)$$\begin{array}{c} f_{=}\left(x,x^{\prime },x^{\prime \prime }\right)=\delta \left(x-x^{\prime }\right)\delta \left(x-x^{\prime \prime }\right) \end{array}$$
to constrain the marginal beliefs over *x*, $x^{\prime }$, $x^{\prime \prime }$ to be equal. With this mechanism, any factorized function can be represented as an FFG.

An FFG visualization of the TVAR model is depicted in Figure 3, but for illustrative purposes, we first consider an example factorized distribution
(17)$$\begin{array}{cc} p(x_{1},x_{2},x_{3},x_{4}) & =p\left(x_{1}\right)p\left(x_{2}|x_{1}\right)p\left(x_{3}|x_{2}\right)p\left(x_{4}|x_{3}\right) \end{array}$$

This distribution can be visualized by an FFG shown in Figure 1. An FFG is in principle an undirected graph but we often draw arrows on the edges in the “generative” direction, which is the direction that describes how the observed data is generated. Assume that we are interested in computing the marginal for $x_{2}$, given by
(18)$$\begin{array}{cc} p(x_{2}) & =\;\int \int \int \;p\left(x_{1},x_{2},x_{3},x_{4}\right)dx_{1}dx_{3}dx_{4} \end{array}$$

We can reduce the complexity of computing this integral by rearranging the factors over the integration signs as
(19a)$$\begin{array}{cc} p(x_{2}) & =\underset{{\vec{\mu }}_{2}\left(x_{2}\right)}{\underset{︸}{\int \underset{{\vec{\mu }}_{1}\left(x_{1}\right)}{\underset{︸}{p(x_{1})}}p\left(x_{2}|x_{1}\right)dx_{1}}}\cdot \underset{{\overset{\leftarrow }{\mu }}_{2}\left(x_{2}\right)}{\underset{︸}{\left(\int p\left(x_{3}|x_{2}\right)\underset{{\overset{\leftarrow }{\mu }}_{3}\left(x_{3}\right)}{\underset{︸}{\left(\int p\left(x_{4}|x_{3}\right)dx_{3})\right)}}dx_{3}\right)}} \end{array}$$
(19b)$$\begin{array}{cc} \; & ={\vec{\mu }}_{2}\left(x_{2}\right)\cdot {\overset{\leftarrow }{\mu }}_{2}\left(x_{2}\right)\;. \end{array}$$

These re-arranged integrals can be interpreted as messages that are passed over the edges, see Figure 1. It is a notational convention to call a message $\vec{\mu }\left(\cdot \right)$ that aligns with the direction of the edge arrow a forward message and similarly, a message $\overset{\leftarrow }{\mu }\left(\cdot \right)$ that opposes the direction of the edge is called a backward message.

This message passed-based algorithm of computing the marginal is called belief propagation (BP) or the sum-product algorithm. As can be verified in (19), for a node with factor $f(y,x_{1},\cdots ,x_{n})$, the outgoing BP message $\vec{\mu }\left(y\right)$ to variable *y* can be expressed as
(20)$${\vec{\mu }}_{y}\left(y\right)=\int \cdots \int f\left(y,x_{1},\cdots ,x_{n}\right)\prod_{i=1}^{n}{\vec{\mu }}_{i}\left(x_{i}\right)dx_{i}\;.$$
where ${\vec{\mu }}_{i}\left(x_{i}\right)$ is an incoming message over edge $x_{i}$. If the factor graph is a tree, meaning that the graph contains no cycles, then BP leads to exact Bayesian inference. A more detailed explanation of belief propagation message passing in FFGs can be found in [26].

### 4.2. Free Energy and Variational Message Passing

Technically, BP is a message passing algorithm that belongs to a family of message passing algorithms that minimize a constrained variational free energy functional [31]. Unfortunately, the sum-product rule (Equation 20) only has a closed-form solution for Gaussian incoming messages ${\vec{\mu }}_{i}\left(x_{i}\right)$ and linear variable relations in $f(y,x_{1},\ldots ,x_{n})$. Another important member of the free energy minimizing algorithms is the Variational Message Passing (VMP) algorithm [22]. VMP enjoys a wider range of analytically computable message update rules.

We shortly review variational Bayesian inference and VMP next. Consider a model p(y,x) with observations y and unobserved (latent) variables x. We are interested in inferring the posterior distribution p(x|y). In variational inference we introduce an approximate posterior q(x) and define a variational free energy functional as
(21)$$\begin{array}{c} F\left[q\right]≜\int q\left(x\right)\log\frac{q(x)}{p(y,x)}dx=\underset{\mathrm{KL}\;\mathrm{divergence}\;D_{\mathrm{KL}}\left(q,p\right)}{\underset{︸}{\int q\left(x\right)\log\frac{q(x)}{p(x|y)}dx}}-\underset{\log-\mathrm{evidence}}{\underset{︸}{\log p(y)}}\;. \end{array}$$

The second term in (Equation 21) (log-evidence) is not a function of the argument of *F*. The first term is a KL-divergence, which is by definition non-negative and only equals zero for q(x)=p(x|y). As a result, variational inference by minimization of F[q] provides
(22)$$q^{*}\left(x\right)=\arg\min_{q}F\left[q\right]$$
which is an approximation to the Bayesian posterior p(x|y). Moreover, the minimized free energy $F[q^{*}]$ is an upper bound for minus log-evidence and in practice is used a model performance criterion. Similarly to (Equation 4), the free energy can be decomposed as
(23)$$F\left[q\right]=\underset{\mathrm{accuracy}}{\underset{︸}{\int q(x)\log p(y|x,m)dx}}-\underset{\mathrm{complexity}}{\underset{︸}{\int q\left(x\right)\log\frac{q(x)}{\underset{\mathrm{prior}}{\underset{︸}{p(x|m)}}}dx}}$$
which underwrites its usage as a performance criterion for model *m*, given observations y.

In an FFG context, the model p(y,x) is represented by a set of connected nodes. Consider a generic node of the FFG, given by $f(y,x_{1},\cdots ,x_{n})$ where in the case of VMP, the incoming messages are approximations to the marginals $q_{i}\left(x_{i}\right),i=1,\cdots ,n$, see Figure 2.

It can be shown that the outgoing VMP message of *f* towards edge *y* is given by [32]
(24)$$\begin{array}{c} \vec{\nu }\left(y\right)\propto \exp\left(\int ...\int \log f\left(y,x_{1},\cdots ,x_{n}\right)\prod_{i=1}q\left(x_{i}\right)dx_{i}\right)\;. \end{array}$$

In this paper, we adopt the notational convention to denote belief propagation messages (computed by (Equation 20)) by μ and VMP messages (computed by (Equation 24)) by ν. The approximate marginal q(y) can be obtained by multiplying incoming and outgoing messages on the edge for *y*
(25)$$\begin{array}{c} q\left(y\right)\propto \vec{\nu }\left(y\right)\overset{\leftarrow }{\nu }\left(y\right)\;. \end{array}$$

This process (compute forward and backward messages for an edge and update the marginal) is executed sequentially and repeatedly for all edges in the graph until convergence. In contrast to BP-based inference, the VMP and marginal update rules (Equation 24) and (Equation 25) lead to closed-form expressions for a large set of conjugate node pairs from the exponential family of distributions. For instance, updating the variance parameter of a Gaussian node with a connected inverse-gamma distribution node results in closed-form VMP updates.

In short, both BP- and VMP-based message passing can be interpreted as minimizing variational free energy, albeit under a different set of local constraints [31]. Typical constraints include factorization and form constraints on the posterior such as $q\left(x\right)=\prod_{i}q_{i}\left(x_{i}\right)$ and q(x)=N(x|μ,Σ), respectively. Since the constraints are local, BP and VMP can be combined in a factor graph to create hybrid message passing-based variational inference algorithms. For a more detailed explanation of VMP in FFGs, we refer to [32]. Note that hybrid message passing does in general not guarantee to minimize variational free energy [33]. However, in our experiments in Section 6 we will show that iterating our stationary solutions by message passing does lead to free energy minimization.

## 5. Variational Message Passing for TVAR Models

In this section, we focus on deriving message passing-based inference in the TVAR model. We develop a TVAR composite factor for the FFG framework and specify the intractable BP messages around the TVAR node. Then we present a message passing-based inference solution.

### 5.1. Message Passing-Based Inference in the TVAR Model

The TVAR model at time step *t* can be represented by an FFG as shown in Figure 3. We are interested in providing a message passing solution to the inference tasks as specified by Equations (Equation 8)–(Equation 14). At the left-hand side of Figure 3, the incoming messages are the priors $p(\theta_{t-1}|y_{1:t-1})$ and $p(x_{t-1}|y_{1:t-1})$. At the bottom of the graph, there is a new observation $y_{t}$. The goal is to pass messages in the graph to compute posteriors $q(\theta_{t}|y_{1:t})$ (message ⑯) and $q(x_{t}|y_{1:t})$ (message ⑪). In order to support smoothing algorithms, we also want to be able to pass incoming prior messages from the right-hand side to outgoing messages ⓭ and ⓲ at the left-hand side. Forward and backward messages are drawn as open and closed circles respectively.

Technically, the generative model (2) at time step *t* for the TVAR model can shortly be written as $p\left(y_{t}|z_{t}\right)p\left(z_{t}|z_{t-1}\right)$, where $z_{t}=\left\{x_{t},\theta_{t},\omega ,\gamma ,\tau \right\}$ are the latent variables. On this view, we can write the free energy functional for the TVAR model at time step *t* as
(26)$$\begin{array}{c} F\left[q\left(z_{t-1},z_{t}|y_{1:t}\right)\right]=\;\int \int \;q\left(z_{t-1},z_{t}|y_{1:t}\right)\log\frac{\overset{\mathrm{posterior}}{\overset{︷}{q(z_{t-1},z_{t}|y_{1:t})}}}{\underset{\mathrm{generative}\;\mathrm{model}}{\underset{︸}{p\left(y_{t}|z_{t}\right)p\left(z_{t}|z_{t-1}\right)}}\underset{\mathrm{prior}\;\mathrm{from}\;\mathrm{past}}{\underset{︸}{p(z_{t-1}|y_{1:t-1})}}}dz_{t-1}dz_{t}\;. \end{array}$$
and minimize F[q] by message passing. In a smoothing context, we would include a prior from the future $p\left(z_{t}|y_{t+1:t+T}\right):=q\left(z_{t}|y_{t+1:t+T}\right)$, yielding a free energy functional
(27)$$F\left[q\left(z_{t-1},z_{t}|y_{1:T}\right)\right]=\;\int \int \;q\left(z_{t-1},z_{t}|y_{1:T}\right)\log\frac{\overset{\mathrm{posterior}}{\overset{︷}{q(z_{t-1},z_{t}|y_{1:T})}}}{\underset{\mathrm{generative}\;\mathrm{model}}{\underset{︸}{p\left(y_{t}|z_{t}\right)p\left(z_{t}|z_{t-1}\right)}}\underset{\begin{array}{c} \mathrm{prior}\\ \mathrm{from}\;\mathrm{past} \end{array}}{\underset{︸}{p(z_{t-1}|y_{1:t-1})}}\underset{\begin{array}{c} \mathrm{prior}\\ \mathrm{from}\;\mathrm{future} \end{array}}{\underset{︸}{p(z_{t}|y_{t+1:t+T})}}}dz_{t-1}dz_{t}.$$

In a filtering context, $q(z_{t}|y_{t+1:t+T})\propto 1$ and the functional (Equation 27) simplifies to (Equation 26).

### 5.2. Intractable Messages and the Composite AR Node

The modularity of message passing in FFGs allows us to focus on only the intractable message and marginal updates. For instance, while there is no problem with the analytical computation of the backward message ⓬, the corresponding forward message ➃,
(28)$$\vec{\mu }\left(x_{t}\right)=\int N\left(x_{t}|A\left(\theta_{t}\right)x_{t-1},V\left(\gamma \right)\right)\underset{\mathrm{Gaussian}\;\mathrm{messages}}{\underset{︸}{\vec{\mu }\left(x_{t-1}\right)\vec{\mu }\left(\theta_{t}\right)\vec{\mu }\left(\gamma \right)}}d\gamma d\theta_{t}x_{t-1}$$
cannot be analytically solved [34]. Similarly, some other messages ⓭, ⓮ and ⓯ do not have a closed-form solution in the constrained free energy minimization framework. For purpose of identification, in Figure 3 intractable messages are marked in red color.

In an FFG framework, we can isolate the problematic part of the TVAR model (Figure 3) by introducing a “composite” AR node. Composite nodes conceal their internal operations from the rest of the graph. As a result, inference can proceed as long as each composite node follows proper message passing communication rules at its interfaces to the rest of the graph. The composite AR node
(29)$$f_{\mathrm{AR}}\left(x_{t},x_{t-1},\theta_{t},\gamma \right)=N\left(x_{t}|A\left(\theta_{t}\right)x_{t-1},V\left(\gamma \right)\right)$$
is indicated in Figure 3 by a dashed box. Note that the internal shuffling of the parameters $\theta_{t}$ and γ, respectively by means of $A(\theta_{t})$ and V(γ), is hidden from the network outside the composite AR node.

### 5.3. VMP Update Rules for the Composite AR Node

We isolate the composite AR node by the specification
(30)$$f_{\mathrm{AR}}\left(y,x,\theta ,\gamma \right)=N\left(y|A(\theta )x,V(\gamma )\right)\;,$$
where, relative to (Equation 29), we used substitutions $y=x_{t},x=x_{t-1},\theta =\theta_{t}$.

Under the structural factorization constraint (See Section A.1 for more on structural VMP).
(31)q(y,x,θ,γ)=q(y,x)q(θ)q(γ),
and consistency constraints
(32)q(y)=∫q(y,x)dx,q(x)=∫q(y,x)dy
the marginals q(θ), q(x), q(y) and q(γ) can be obtained from the minimisation of the composite-AR free energy functional
(33)$$\begin{array}{cc} F_{\mathrm{AR}}\left[q\right] & =\int q\left(y,x\right)q\left(\theta \right)q\left(\gamma \right)\log\frac{\overset{\mathrm{posterior}}{\overset{︷}{q(y,x)q(\theta )q(\gamma )}}}{\underset{\mathrm{AR}\;\mathrm{node}}{\underset{︸}{f_{\mathrm{AR}}\left(y,x,\theta ,\gamma \right)}}}dydxd\theta d\gamma \;. \end{array}$$

Recalling (Equation 25), we can write the minimizer of FE functional (Equation 33) with respect to θ as
(34)$$q\left(\theta \right)\propto \vec{\nu }\left(\theta \right)\overset{\leftarrow }{\nu }\left(\theta \right)$$
where q(θ) is associated with the incoming message to AR node and $\vec{\nu }\left(\theta \right)$ is a variational outgoing message. Hence, the outgoing message from the AR node toward θ can be written as
(35)$$\begin{array}{cc} \vec{\nu }\left(\theta \right) & \propto \exp\left(E_{q\left(y,x\right)q\left(\theta_{t}\right)q\left(\gamma \right)}\log\left[N\left(y|\mathrm{Ax},V\right)\right]\right) \end{array}$$

In Appendix A we work out a closed-form solution for this and all other update rules plus an evaluation of free energy for the composite AR node. The results are reported in Table 1. With these rules in hand, the composite AR node can be plugged into any factor graph and take part in a message passing-based free energy minimization process.

## 6. Experiments

In this section, we first verify the proposed methodology by a simulation of the proposed TVAR model on synthetic data, followed by validation experiments on two real world problems. We implemented all derived message passing rules in the open source Julia package ForneyLab.jl [28]. The code for the experiments and for the AR node can be found in public Github repositories. (https://github.com/biaslab/TVAR_FFG, accessed on 27 May 2021, https://github.com/biaslab/LAR, accessed on 27 May 2021) We used the following computer configuration to run the experiments. *Operation system*: macOS Big Sur, *Processor*: 2,7 GHz Quad-Core Intel Core i7, *RAM*: 16 GB.

### 6.1. Verification on a Synthetic Data Set

To verify the proposed TVAR inference methods, we synthesized data from two generative models $m_{1}$ and $m_{2}$, as follows:
(36a)$$\begin{array}{cc} \theta_{t} & ∼\left\{\begin{array}{cc} \delta (\theta_{t}-\theta_{t-1}) & \mathrm{if}\;m=m_{1}\\ N(\theta_{t-1},\omega I_{M}) & \mathrm{if}\;m=m_{2} \end{array}\right. \end{array}$$
(36b)$$\begin{array}{cc} x_{t} & ∼N\left(A\left(\theta_{t}\right)x_{t-1},V\left(\gamma \right)\right) \end{array}$$
(36c)$$\begin{array}{cc} y_{t} & ∼N(c^{T}x_{t},\tau ) \end{array}$$
with priors
(37a)$$\begin{array}{cc} p(M=k) & =\prod_{k=1}^{10}0.1^{\left[M=k\right]} \end{array}$$
(37b)$$\begin{array}{cc} \theta_{0} & ∼\left\{\begin{array}{cc} N(0,I) & \mathrm{if}\;m=m_{1}\\ N(0,1e12I) & \mathrm{if}\;m=m_{2} \end{array}\right. \end{array}$$
(37c)$$\begin{array}{cc} x_{0} & ∼N(0,1e12I) \end{array}$$
(37d)$$\begin{array}{cc} \gamma & ∼\Gamma (1.0,1e-5) \end{array}$$
(37e)$$\begin{array}{cc} \tau & =1.0 \end{array}$$
(37f)$$\begin{array}{cc} \omega & =0.01 \end{array}$$
where *M* is the number of AR coefficients. Although these models differ only with respect to the properties of the AR coefficients θ, this variation has an important influence on the data generative process. The first model $m_{1}$ specifies a stationary AR process, since $\delta (\theta_{t}-\theta_{t-1})$ in ([Disp-formula FD36a-entropy-23-00683]) indicates that θ is not time-varying in $m_{1}$. The second model $m_{2}$ represents a proper TVAR process as the prior evolution of the AR coefficients follows a random walk. One-time segment FFGs corresponding to the Equation (36) are depicted in Figure 4.

For each model, we generated a data set of 100 different time series, each of length 100 (so we have 2×100×100 data points). For each time series, as indicated by ([Disp-formula FD37a-entropy-23-00683]), the AR order *M* of the generative process was randomly drawn from the set {1,2,…,10}. We used rather non-informative/broad priors for states and parameters for both models, see (37). This was done to ensure that the effect of the prior distributions is negligible relative to the information in the data set.

These time series were used in the following experiments. We selected two recognition models $m_{1}$ and $m_{2}$ with the same specifications as were used for generating the data set. The recognition models were trained on time series that were generated by models with the same AR order.

We proceeded by computing the quantities $q(x_{1:T}|y_{1:T})$, $q(\theta_{1:T}|y_{1:T})$, $q(\gamma |y_{1:T})$ and $F[q\left(z_{t-1},z_{t}|y_{1:T}\right)]$ (where z comprises all latent states and parameters) for both models, following the proposed rules from Table 1.

As a verification check, we first want to ensure that inference recovers the hidden states $x_{t}$ for each t∈(1,2,⋯,100). Secondly, we want to verify the convergence of FE. As we have not used any approximations along the derivations of variational messages, we expect a smoothly decreasing curve for FE until convergence. The results of the verification stage are highlighted for a typical case in Figure 5. The figure confirms that states $x_{t}$ are accurately tracked and that a sliding average of the AR coefficients $\theta_{t}$ is also nicely tracked. Figure 5 also indicates that the FE uniformly decreases towards lower values as we spend more computational power.

We note that the FE score by itself does not explain whether the model is good or not, but it serves as a good measure for model comparison. In the following subsection, we demonstrate how FE scores can be used for model selection.

### 6.2. Temperature Modeling

AR models are well-known for predicting different weather conditions such as wind, temperature, precipitation, etc. Here, we will revisit the problem of modeling daily temperature. We used a data set of daily minimum temperatures (in °C) in Melbourne, Australia, 1981–1990 (3287 days) (https://www.kaggle.com/paulbrabban/daily-minimum-temperatures-in-melbourne, accessed on 27 May 2021). We then corrupted the data set by adding random noise sampled from N(0,10.0) to the actual temperatures. A fragment of the time-series is depicted in Figure 6.

To estimate the actual temperature based on past noisy observations by computing $q(x_{t}|y_{1:t})$, we use a TVAR model with measurement noise to simulate uncertainty about corrupted observations. The model is specified by the following equation set
(38a)$$\begin{array}{cc} \theta_{t} & ∼N(\theta_{t-1},I_{M}) \end{array}$$
(38b)$$\begin{array}{cc} x_{t} & ∼N\left(A\left(\theta_{t}\right)x_{t-1}+c\eta ,V\left(\gamma \right)\right) \end{array}$$
(38c)$$\begin{array}{cc} y_{t} & ∼N(c^{⊺}x_{t},\tau ) \end{array}$$
with priors
(39a)$$\begin{array}{cc} \theta_{0}∼N(0, & I)\;x_{0}∼N\left(0,I\right)\;\eta ∼N\left(0.0,10.0\right) \end{array}$$
(39b)$$\begin{array}{cc} \; & \gamma ∼\Gamma (1.0,1.0)\;\tau ∼\Gamma (0.1,1.0) \end{array}$$

Since the temperature data is not centered around 0 °C, we added a bias term η to the state $x_{t}$. The corresponding FFG is depicted in Figure 7.

Note that we put a Gamma prior on the measurement noise precision τ, meaning that we are uncertain about the size of the error of the thermometer reading. The inference task for the model is computing $q(x_{t}|y_{1:t})$, in other words, we track the states based only on past data. Of course, after training, we could use the model for temperature prediction by tracking $q(x_{t+k}|y_{1:t})$ for k≥1. We compare the performance of four TVAR models with AR orders M={1,2,3,4}. To choose the best model, we computed the average FE score for each TVAR(*M*) model.

Figure 8 shows that on average TVAR(3) outperforms its competitors. The complexity vs accuracy decomposition (Equation 23) of FE explains why a lower order model may outperform higher order models. TVAR(4) maybe as accurate or more accurate than TVAR(3) but the increase in accuracy is more than offset by the increase in complexity. For the lower order models, it is the other way around: they are less complex and involve fewer computations than TVAR(3), but the loss in model complexity leads to too much loss in data modeling accuracy. Overall, TVAR(3) is the best model for this data set. Practically, we always favor the model that features the lowest FE score. In the next subsection we will use this technique (scoring FE) for online model selection.

### 6.3. Single-Channel Speech Enhancement

Single-channel speech enhancement (SCSE) is a well-known challenging task that aims to enhance noisy speech signals that were recorded by a single microphone. In single microphone recordings, we cannot use any spatial information that is commonly used in beamforming applications. Much work has been done to solve the SCSE task, ranging from Wiener filter-inspired signal processing techniques [35,36] to deep learning neural networks [37]. In this paper, we use data from the speech corpus (NOIZEUS) (https://ecs.utdallas.edu/loizou/speech/noizeus/, accessed on 27 May 2021) [38] and corrupted clean speech signals with white Gaussian noise, leading to a signal-to-noise ratio (SNR)
(40)$$\mathrm{SNR}\left(s_{1:T},y_{1:T}\right)=10\log_{10}\left[\frac{\sum_{t}^{T}s_{t}^{2}}{\sum_{t}^{T}{\left(s_{t}-y_{t}\right)}^{2}}\right]\approx 13.36\;\mathrm{dB}$$
where $s_{1:T}=\left(s_{1},\cdots ,s_{T}\right)$ and $y_{1:T}=\left(y_{1},\cdots ,y_{T}\right)$ are clean and corrupted speech signals. $s_{t}$ is a speech signal at time *t* and *T* is the length of the signal.

Historically, AR models have shown to perform well for modeling speech signals in the time (waveform) domain [39,40]. Despite the fact that speech is a highly nonstationary signal, we may assume it to be stationary within short time intervals (frames) of about 10 [ms] each [41]. Since voiced, unvoiced and silence frames have very different characteristics, we used 5 different models (a random walk model (RW), AR(1), AR(2), TVAR(1) and TVAR(2)) for each frame of 10 [ms] with 2.5 [ms] overlap. Given a sampling frequency of 8 [kHZ], each frame results into 80 samples with 20 samples overlap. The AR and TVAR models were specified by Equation (36). For each frame, we evaluated the model performance by minimized FE and selected the model with minimal FE score. We used identical prior parameters for all models where possible. To recover the speech signal we computed the mean values of $q(x_{t}|y_{1:T})$ of the selected model for each frame. The SNR gain of this SCSE system was
(41)$$\mathrm{SNR}\left(s_{1:T},x_{1:T}\right)-\mathrm{SNR}\left(s_{1:T},y_{1:T}\right)\approx 3.7\;\mathrm{dB}.$$

Figure 9 show the spectrograms of the clean, noisy and filtered signal respectively.

Next, we analyze the inference results in a bit more detail. Table 2 shows the percentage of winning models for each frame based on the free energy score.

As we can see, for more than 30% of all frames, the random walk model performs best. This happens mostly because for a silent frame the AR model gets penalized by its complexity term. We recognize that in about 90% of the frames the best models are AR(1) and RW. On the other hand, for the frames where the speech signal transitions from silent or unvoiced to voiced, these fixed models start to fail and the time-varying AR models perform better. This effect can be seen in Figure 10.

Figure 11 shows the performance of the AR(2) and RW models on a frame with a voiced speech signal. For this case, the AR(2) model performs better.

Finally, Figure 12 shows how the TVAR(2) model compares to the RW model on one of the unvoiced/silence frames. While the estimates of TVAR(2) appear to be more accurate, it pays a bigger “price” for the model complexity term in the FE score and the RW model wins the frame.

## 7. Discussion

We have introduced a TVAR model that includes efficient joint variational Bayesian tracking of states, parameters and free energy. The system can be used as a plug-in module in factor graph-based representations of other models. At several points in this paper, we have made some design decisions that we shortly review here.

While FE computation for the AR node provides a convenient performance criterion for model selection, we noticed in the speech enhancement simulation that separate FE tracking for each candidate model leads to a large computational overhead. There are ways to improve the model selection process that we used in the speech enhancement simulation. One way is to consider a mixture model of candidate models and track the posterior over the mixture coefficients [42]. Alternatively, a very cheap method for online Bayesian model selection may be the recently developed Bayesian Model Reduction (BMR) method [43]. The BMR method is based on a generalization of the Savage-Dickey Density Ratio and supports tracking of free energy of multiple nested models with almost no computational overhead. Both methods seem to integrate well with a factor graph representation and we plan to study this issue in future work.

In this paper, the posterior factorization (Equation 31) supports the modeling of temporal dependencies between input and output of the AR node in the posterior. Technically, (Equation 31) corresponds to a structural VMP assumption, in contrast to the more constrained mean-field VMP algorithm that would be based on $q\left(z\right)=\prod_{i}q_{i}\left(z_{i}\right)$, where z is the set of all latent variables [44]. We could have also worked out alternative update rules for the assumption of a joint factorization of precision γ and AR coefficients θ. In that case, the prior (incoming message $\vec{\nu }\left(\theta ,\gamma \right)$ to AR node) would be in the form of a Normal-Gamma distribution. While any of these these assumptions are technically valid, each choice accepts a different trade-off in the accuracy vs. complexity space. We review structural VMP in Section A.1.

In the temperature modelling task, we added some additional random variables (bias, measurement noise precision). To avoid identifiability issues, in ([Disp-formula FD38a-entropy-23-00683]) we fixed the covariance matrix of the time-varying AR coefficient to the identity matrix. In principle, this constraint can be relaxed. For example, an Inverse-Wishart prior distribution can be added to the covariance matrix.

In our speech enhancement experiments in Section 6.3, we assume that the measurement noise variance is known. In a real-world scenario, this information is usually not accessible. However, online tracking of measurement noise or other (hyper-)parameters is usually not a difficult extension when the process is simulated in a factor graph toolbox such as ForneyLab [28]. If so desired, we could add a prior on the measurement noise variance and track the posterior. The online free energy criterion (Equation 23) can be used to determine if the additional computational load (complexity) of Bayesian tracking of the variance parameter has been compensated by the increase in modeling accuracy.

The realization of the TVAR model in ForneyLab comes with some limitations. For large smoothing problems (say, >1000 data points), the computational load of message passing in ForneyLab becomes too heavy for a standard laptop (as was used in the paper). Consequently, in the current implementation it is difficult to employ the AR node for processing large time series on a standard laptop. To circumvent this issue, when using ForneyLab, one can combine filtering and smoothing solutions into a batch learning procedure. In future work we plan to remedy this issue by some ForneyLab refactoring work. Additionally, the implemented AR node does not provide a closed-form update rule for the marginal distribution when the probability distribution types of the incoming messages (priors) are different from the ones used in our work. Fortunately, ForneyLab supports resorting to (slower) sampling-based update rules when closed-form update rules are not available.

## 8. Conclusions

We presented a variational message passing approach to tracking states and parameters in latent TVAR models. The required update rules have been summarized and implemented in the factor graph package ForneyLab.jl, thus making transparent usage of TVAR factors available in freely definable stochastic dynamical systems. Aside from VMP update rules, we derived a closed-form expression for the variational free energy (FE) of an AR factor. Free Energy can be used as a proxy for Bayesian model evidence and as such allows for model performance comparisons between the TVAR models and alternative structures. Owing to the locality and modularity of the FFG framework, we demonstrated how AR nodes can be applied as plug-in modules in various dynamic models. We verified the correctness of the rules on a synthetic data set and applied the proposed TVAR model to a few relatively simple but different real-world problems. In future work, we plan to extend the current factor graph-based framework to efficient and transparent tracking of AR model order and to online model comparison and selection with alternative models.

## Figures and Tables

**Figure 1 entropy-23-00683-f001:**
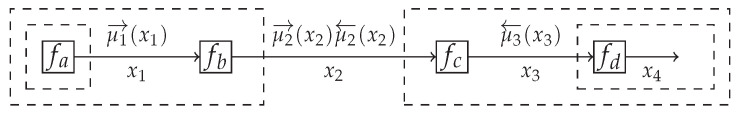
An FFG corresponding to model (Equation 17), including messages as per (19). For graphical clarity, we defined $f_{a}\left(x_{1}\right)=p\left(x_{1}\right)$, $f_{b}\left(x_{1},x_{2}\right)=p\left(x_{2}|x_{1}\right)$, $f_{c}\left(x_{2},x_{3}\right)=p\left(x_{3}|x_{2}\right)$ and $f_{d}\left(x_{3},x_{4}\right)=p\left(x_{4}|x_{3}\right)$.

**Figure 2 entropy-23-00683-f002:**
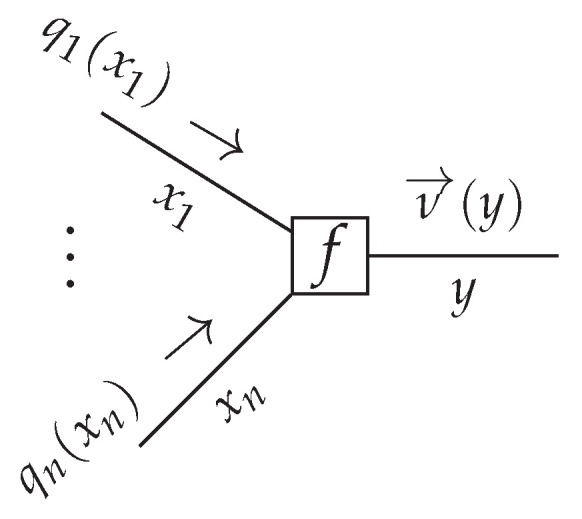
A generic node $f(y,x_{1},\cdots ,x_{n})$ with incoming variational messages $q_{i}\left(x_{i}\right)$ and outgoing variational message $\vec{\nu }\left(y\right)$, see Equation (Equation 24). Note that the marginals q(·) propagate in the graph as messages.

**Figure 3 entropy-23-00683-f003:**
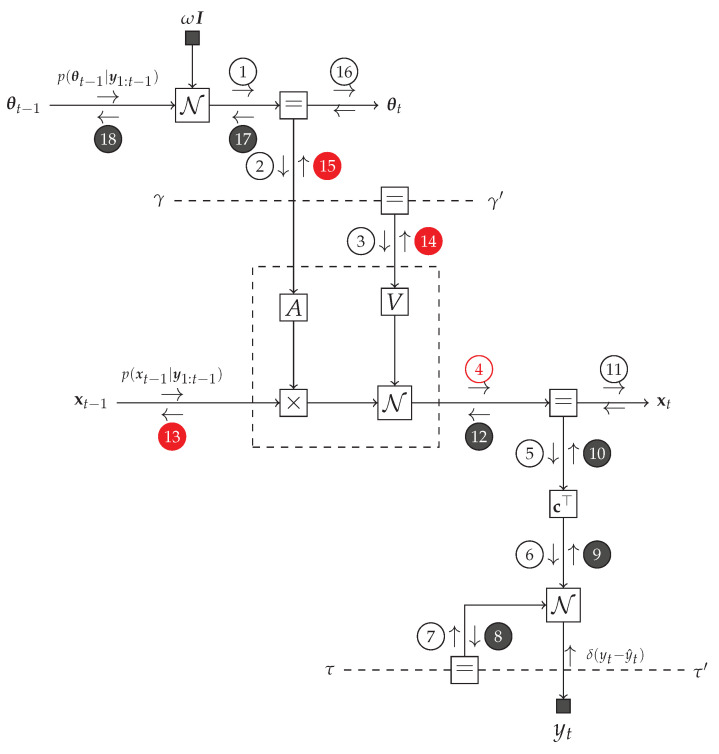
One time segment of an FFG corresponding to the TVAR model. We use small black nodes to denote observations and fixed given parameter values. The observation node for $y_{t}$ sends a message $\delta (y_{t}-{\hat{y}}_{t})$ into the graph to indicate that $y_{t}={\hat{y}}_{t}$ has been observed. Dashed undirected edges denote time-invariant variables. Circled numbers indicate a selected computation schedule. Backward messages are marked by black circles. The intractable messages are labeled with red. The dashed box represents a composite AR node as specified by (Equation 30).

**Figure 4 entropy-23-00683-f004:**
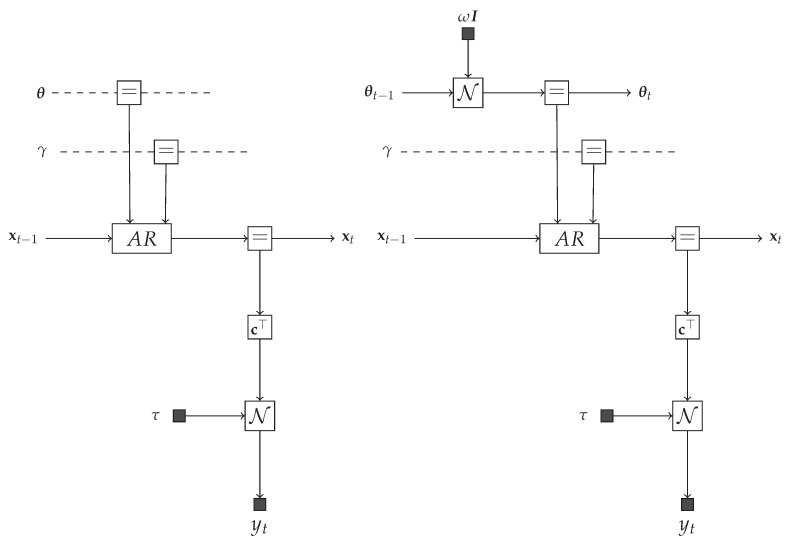
Forney-style Factor Graphs corresponding to Equation (36). (**Left**) model $m_{1}$. (**Right**) model $m_{2}$.

**Figure 5 entropy-23-00683-f005:**
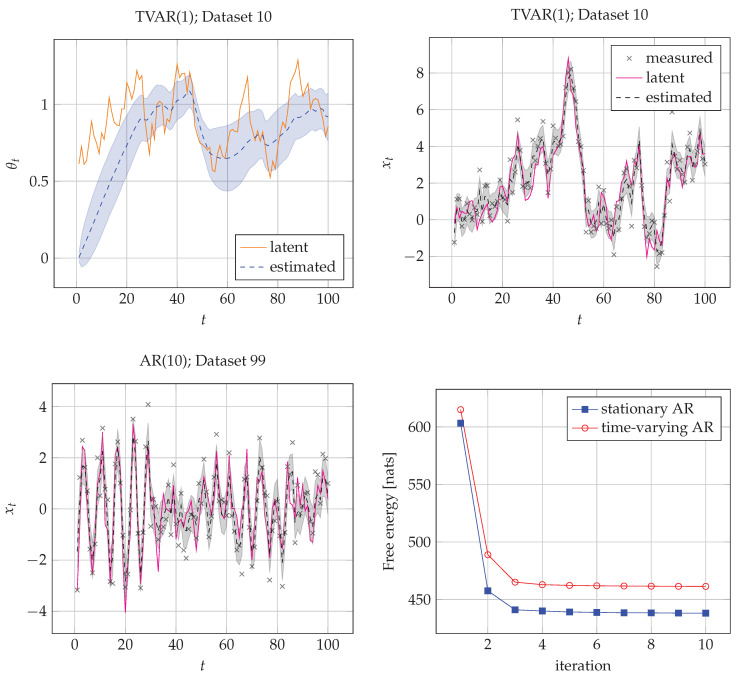
Verification results. The solid line corresponds to the value of the latent (hidden) states in the generative processes. The dashed line corresponds to the expected mean value of the posterior estimates of hidden states $q(\cdot |y_{1:100})$ in the recognition models. The shadowed regions corresponds to one standard deviation of the posteriors in the recognition models below and above the estimated mean. The top two plots show inference results for the coefficients $\theta_{t}$ (top-left) and states $x_{t}$ (top-right) of TVAR(1) (model $m_{2}$, AR order M=1) for time series ♯10. (bottom-left) State trajectory $q(x_{t}|y_{1:100})$ model $m_{1}$, AR order M=1 on time series ♯99. (Bottom-right) Evolution of FE for $m_{1}$ (AR) and $m_{2}$ (TVAR), averaged over their corresponding time series. The iteration number at the abscissa steps through a single marginal update for all edges in the graph.

**Figure 6 entropy-23-00683-f006:**
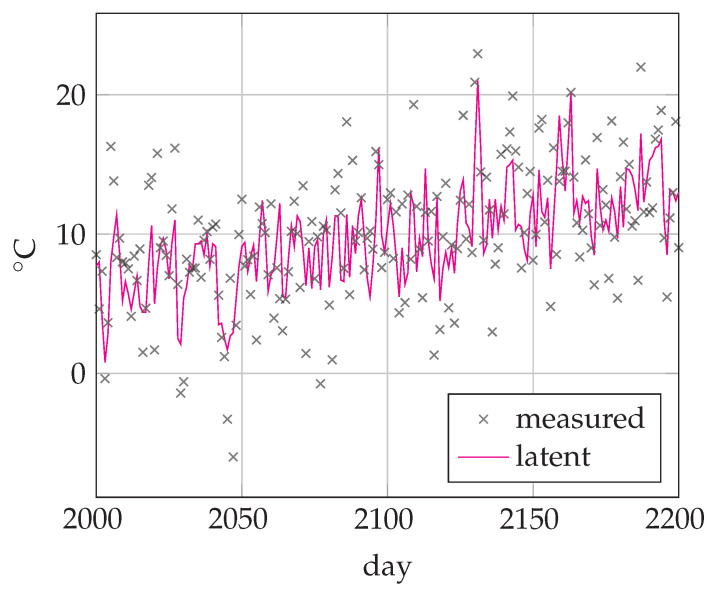
Temperature time-series from days 2000 to 2200. Crosses denote the thermometer readings plus added noise. The solid line corresponds to the latent (hidden) daily temperature.

**Figure 7 entropy-23-00683-f007:**
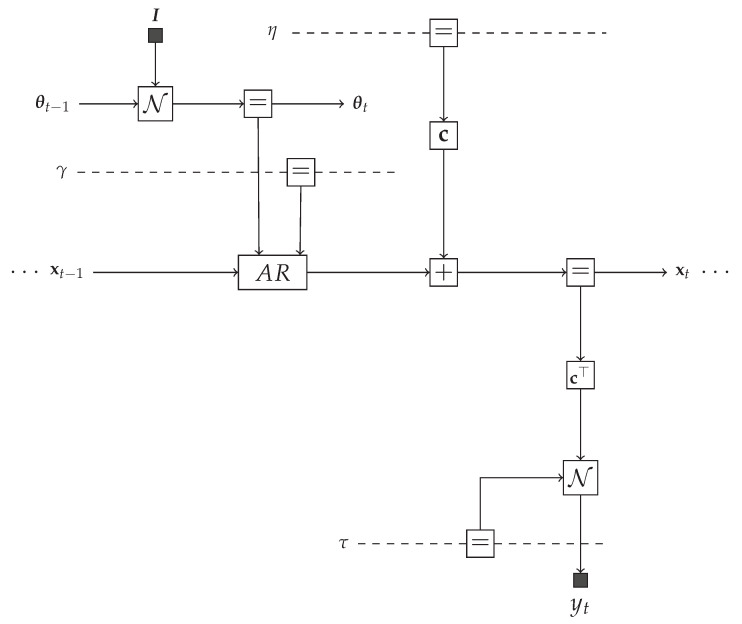
One time segment of a Forney-style factor graph (FFG) for the TVAR model in the temperature modeling task (38).

**Figure 8 entropy-23-00683-f008:**
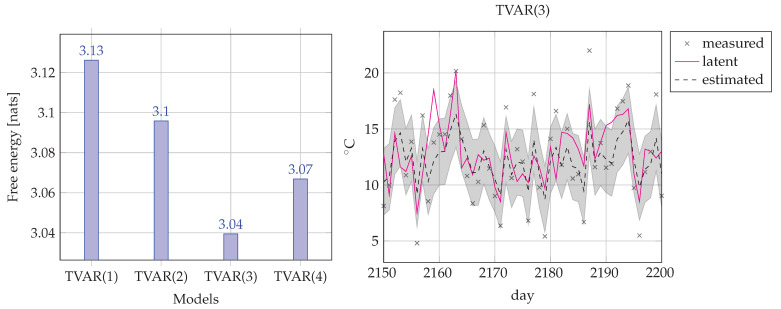
(**Left**) Comparison of four TVAR(*M*) models for the temperature filtering problem. Bars correspond to the averaged (over 3287 days) FE score for each model. (**Right**) Inference example of the best performing model (TVAR(3)). Crosses denote the thermometer reading plus added noise. The solid line corresponds to the latent (hidden) daily temperature. The dashed line corresponds to the mean of the posterior estimates of hidden temperature and the shadowed region corresponds to one standard deviation below and above the estimated temperature.

**Figure 9 entropy-23-00683-f009:**
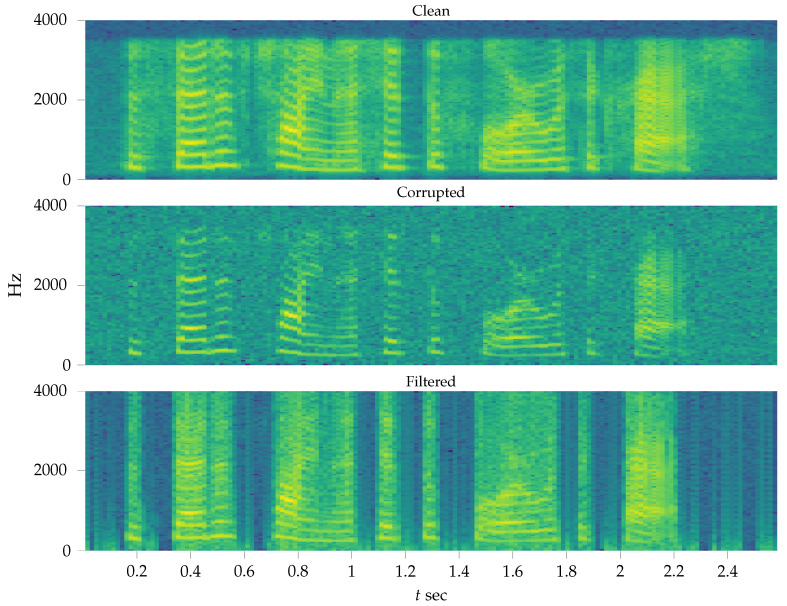
Spectrogram of recovered speech signal in the experiment of Section 6.3.

**Figure 10 entropy-23-00683-f010:**
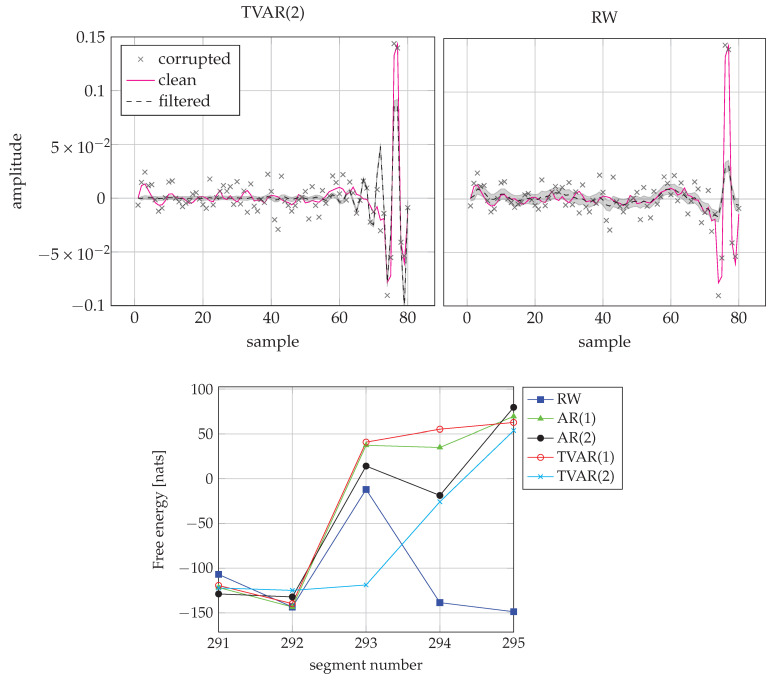
(Top) (**Top-left**) Inference by TVAR(2) for the segment 293. (**Top-right**) Inference by RW for the segment 293. Note how the TVAR model is able to follow the transitions at the end of the frame, while the RW cannot adapt within one frame. (**Bottom**) FE scores from segment 291 to 295. TVAR(2) wins frame 293 as it has the lowest FE score.

**Figure 11 entropy-23-00683-f011:**
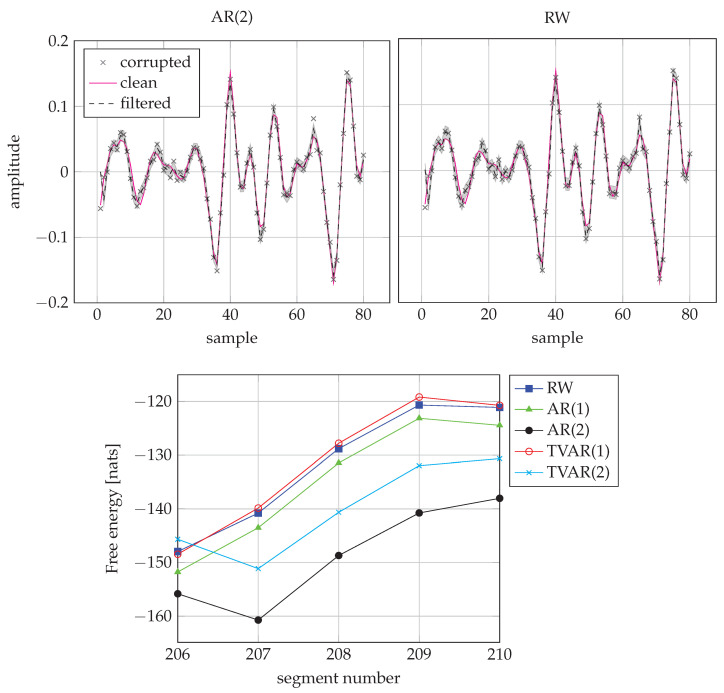
Comparison of AR(2) and RW models for a voiced signal frame. (**Top-left**) Inference by AR(2) for the segment 208. (**Top-right**) Inference by RW for the segment 208. (**Bottom**) FE scores from segment 206 to 210. The AR(2) model wins frame 208.

**Figure 12 entropy-23-00683-f012:**
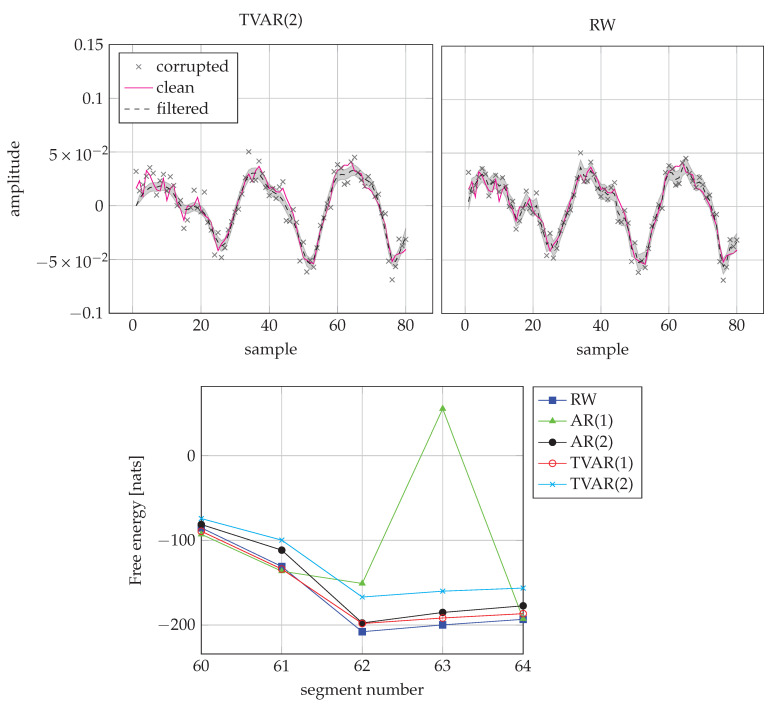
Comparison of TVAR(2) and RW models for an unvoiced/silence frame. (**Top-left**) Inference by TVAR(2) for the frame 62. (**Top-right**) Inference by RW for the frame 62. **(Bottom)** FE scores from segment 60 to 64. The RW model scores best on frame 62 due to its low complexity.

**Table 1 entropy-23-00683-t001:** Variational message update rules for the autoregressive (AR) node (dashed box) of Equation (Equation 30).

VMP for the Composite AR Node	
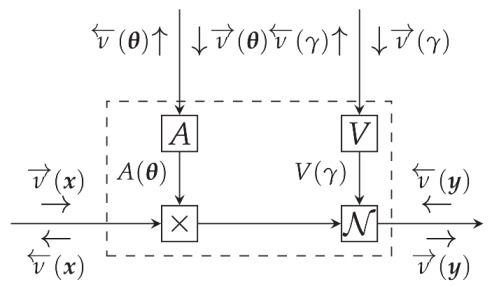	
**Outgoing messages**	**Incoming messages**
$\;\begin{array}{cc} \;\;\; & \vec{\nu }\left(y\right)\propto N\left(y|z_{0},\Sigma \right)\;\;\;\\ & \overset{\leftarrow }{\nu }\left(x\right)\propto N\left(x|\Lambda_{1}^{-1}z_{1},\Lambda_{1}^{-1}\right)\\ & \overset{\leftarrow }{\nu }\left(\theta \right)\propto N\left(\theta |\Lambda_{2}^{-1}z_{2},\Lambda_{2}^{-1}\right)\\ & \overset{\leftarrow }{\nu }\left(\gamma \right)\propto \Gamma \left(\gamma |\frac{3}{2},\frac{b}{2}\right) \end{array}$	$\;\begin{array}{cc} & \overset{\leftarrow }{\nu }\left(y\right)\propto N\left(y|m_{y},V_{y}\right)\\ & \vec{\nu }\left(x\right)\propto N\left(x|m_{x},V_{x}\right)\\ & \vec{\nu }\left(\theta \right)\propto N\left(\theta |m_{\theta },V_{\theta }\right)\\ & \vec{\nu }\left(\gamma \right)\propto \Gamma \left(\gamma |\alpha ,\beta \right) \end{array}$
**Joint marginal q(y,x)**	
$q\left(y,x\right)\propto \vec{\nu }\left(x\right)\exp\left[E_{q(\gamma )q(\theta )}\log f\left(y\;\;x,\theta ,\gamma \right)\right]\overset{\leftarrow }{\nu }\left(y\right)\propto N\left(\left[\begin{array}{c} y\\ x \end{array}\right]|\left[\begin{array}{c} m_{y}^{*}\\ m_{x}^{*} \end{array}\right],\left[\begin{array}{cc} V_{y}^{*} & V_{\mathrm{yx}}^{*}\\ V_{\mathrm{xy}}^{*} & V_{x}^{*} \end{array}\right]\right)$ (Section A.7)	
**Free energy** F[q]	
$\begin{array}{c} F\left[q\right]=u+\frac{1}{2}\log2\pi +\frac{m_{\gamma }}{2}\left(\sigma_{y}^{2}+m_{y}^{2}-2\left[V_{yx^{⊺}}+m_{y}m_{x}^{⊺}\right]m_{\theta }+\mathrm{tr}\left[\left(V_{\theta }+m_{\theta }m_{\theta }^{⊺}\right)V_{x}^{*}\right]+m_{\theta }^{⊺}\left(V_{x}^{*}+m_{x}^{*}{\left(m_{x}^{*}\right)}^{⊺}\right)m_{\theta }\right) \end{array}$	
**Auxiliary variables**	
$\begin{array}{cc} & \Sigma =\;m_{A}{\left(V_{x}^{-1}+m_{\gamma }V_{\theta }\right)}^{-1}m_{A}^{⊺}+m_{V}\;\;\;\;z_{0}=m_{A}{\left(V_{x}^{-1}+m_{\gamma }V_{\theta }\right)}^{-1}V_{x}^{-1}m_{x}\\ & \Lambda_{1}=\;m_{A}^{⊺}{\left(V_{y}+m_{V}\right)}^{-1}m_{A}+m_{\gamma }V_{\theta }\;\;\;z_{1}=m_{A}^{⊺}{\left(V_{y}+m_{V}\right)}^{-1}m_{y}\\ & \Lambda_{2}=\;m_{\gamma }\left(V_{x}^{*}+m_{x}^{*}{\left(m_{x}^{*}\right)}^{⊺}\right)\;\;\;\;\;z_{2}=\left(V_{\mathrm{xy}}^{*}+m_{x}^{*}{\left(m_{y}^{*}\right)}^{⊺}\right)cm_{\gamma }\\ & b=\;c^{⊺}\left[V_{y}^{*}+m_{y}^{*}{\left(m_{y}^{*}\right)}^{⊺}-2m_{A}\left(V_{\mathrm{xy}}^{*}+m_{x}^{*}{\left(m_{y}^{*}\right)}^{⊺}\right)+m_{A}\left(V_{x}^{*}+m_{x}^{*}{\left(m_{x}^{*}\right)}^{⊺}\right)m_{A}^{⊺}+\mathrm{tr}\left(V_{\theta }\left(V_{x}^{*}+m_{x}^{*}{\left(m_{x}^{*}\right)}^{⊺}\right)\right)\right]c\\ & m_{\gamma }=\frac{\alpha }{\beta }\;m_{A}=m_{A(\theta )}\\ & u=-\frac{1}{2}\left[\psi (\alpha )-\log\beta \right]+\frac{1}{2}\log2\pi \;\sigma_{y}^{2}=c^{⊺}V_{y}^{*}c\;m_{y}=c^{⊺}m_{y}^{*}c\;V_{yx}=V_{\mathrm{yx}}^{*}c \end{array}$	

**Table 2 entropy-23-00683-t002:** Percentage of preferred models (based on FE scores) for all frames on the speech enhancement task.

	RW	AR(1)	AR(2)	TVAR(1)	TVAR(2)
Ratio	32.2%	54.3%	10.7%	1.2%	0.5%

## Data Availability

Not applicable.

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
