# Peer review of "Message Passing-Based Inference for Time-Varying Autoregressive Models"

_entropy, 2021, doi:10.3390/e23060683_

Round 1
Reviewer 1 Report
Report on the manuscript "entropy-1207516" entitled "Message Passing-based Inference for Time-Varying Autoregressive Models"
This manuscript proposes structured variational update rules for a composite autoregressive node with probabilistic observations to model the time-varying behavior of the hyper-parameters of a time-varying autoregressive model. Some experimental results are presented to illustrate the proposed methods based on synthetic data set and validated with real-world data from temperature modeling and speech enhancement tasks. A discussion and conclusions about the present investigation are reported.
In general, I have a good opinion about this work and recommend its acceptance after a minor revision that considers the following concerns:
1. The manuscript needs to be proofread by the author carefully. Just as an example, the abstract says "rules for an composite AR node".
2. Words in the title are not usually in the keywords. In addition, the keywords are often written in alphabetical order.
3. In the abstract, some general ideas about the obtained conclusions should be added.
4. To the best of my knowledge, there has been a good body of work done in the literature on this topic. The novelty and contribution of this study must be clearly stated.
5. The authors must check the use of all acronyms and abbreviations employed in the whole manuscript. Please avoid to use "i.e.", "e.g.", etcetera. Instead, "that is" and "for example" can be used, respectively.
6. Notations and mathematical symbols must be checked and fixed.
7. In my opinion, the implications of the study are underdeveloped and must be explained further in the conclusions.
8. The authors must provide more details about the computational framework used in the manuscript. For example, software and packages used, features of the computer employed, and other computational aspects must be added. In addition, I recommend summarizing the methodology in an algorithm so that the readers can follow it easier.
9. I do not have checked each formula or numerical results in detail. I recommend the author to check them.
10. The final conclusion needs to be improved. The author must add limitations of the study and more ideas for further research. Then, I suggest titling the final section as "Conclusions, limitations, and future research".
11. The author must check whether all references are cited and whether all citations are in the reference list, as well as making an effort to discuss and cite any paper on the topic published in Entropy to attract the attention of our target audience.
12:
Sections and subsections are not well organized. Some sections have subsections and others not, some have an introduction and others not (see Sections 3 and 6), even a section has subsubsections (see Section 6). The authors must look for a manner to organize the manuscript so that its readability and presentation are improved.Author Response
Dear reviewer,
Please find our reply in the attachment.
Best regards,
Albert Podusenko

Reviewer 2 Report
This paper presents a Bayesian approach for Time-Varying Autoregressive Models. The Bayesian approach is based on a variational message passing approach over factor graphs.
The addressed problem is relevant because TVAR models can be applied to a wide range of problems. Purse a Bayesian approach is well motivated because it allows to capture model uncertainty and provide well-founded methods for scoring models.
In general, the paper is very well-written and easy to follow. The concepts and the methods are clearly introduced and discussed. The authors also provide a good revision of the literature.
The experimental evaluation is well performed on artificial and relevant real-world data sets and the results are conclusive.
In general, this work does not present a strong contribution. The presented methods (VMP + Factor Graphs) are widely known in the literature. But this work has properly applied them to a challenging and relevant problem, as it is the case of TVAR models.
Minor Comments:
Line 39. "on ultra-low power platforms" --> I would tune down this claim because it has not been really evaluated.
Equations 8-11. Some subscripts are in bold.
Eq (23). It would be nice to comment that p(x|m) acts as a prior.
Line 141. Incoming messages in VMP are "approximations to the marginals" not marginals.
Line 152. I think it is important to note that hybrid schemes do not minimize the free variational energy function. Or, do you mean that some messages really passed the true marginal because the variational approximation is exact in these cases? This should be clarified.
Eq (38a). You are fixing omega=1. Why? Actually, the role of this parameter in your model seems to be very relevant. Some discussion should be included.
Author Response
Dear reviewer,
Please find our reply in the attachment.
Best regards,
Albert Podusenko
